# Single-cell characterization of human GBM reveals regional differences in tumor-infiltrating leukocyte activation

Philip Schmassmann[1]*, Julien Roux[2,3], Steffen Dettling[4], Sabrina Hogan[1], Tala Shekarian[1], Tomás A Martins[1], Marie-Françoise Ritz[1], Sylvia Herter[5], Marina Bacac[5], Gregor Hutter[1,6]*

[1]Brain Tumor Immunotherapy Lab, Department of Biomedicine, University of Basel, Basel, Switzerland; [2]Bioinformatics Core Facility, Department of Biomedicine, University of Basel, Basel, Switzerland; [3]Swiss Institute of Bioinformatics, Basel, Switzerland; [4]Roche Pharmaceutical Research and Early Development, Roche Innovation Center Munich, Penzberg, Germany; [5]Roche Pharmaceutical Research and Early Development, Roche Innovation Center Zürich, Schlieren, Switzerland; [6]Department of Neurosurgery, University Hospital Basel, Basel, Switzerland

*For correspondence:
p.schmassmann@unibas.ch (PS);
gregor.hutter@usb.ch (GH)

**Abstract** Glioblastoma (GBM) harbors a highly immunosuppressive tumor microenvironment (TME) which influences glioma growth. Major efforts have been undertaken to describe the TME on a single-cell level. However, human data on regional differences within the TME remain scarce. Here, we performed high-depth single-cell RNA sequencing (scRNAseq) on paired biopsies from the tumor center, peripheral infiltration zone and blood of five primary GBM patients. Through analysis of >45,000 cells, we revealed a regionally distinct transcription profile of microglia (MG) and monocyte-derived macrophages (MdMs) and an impaired activation signature in the tumor-peripheral cytotoxic-cell compartment. Comparing tumor-infiltrating CD8+ T cells with circulating cells identified CX3CR1high and CX3CR1int CD8+ T cells with effector and memory phenotype, respectively, enriched in blood but absent in the TME. Tumor CD8+ T cells displayed a tissue-resident memory phenotype with dysfunctional features. Our analysis provides a regionally resolved mapping of transcriptional states in GBM-associated leukocytes, serving as an additional asset in the effort towards novel therapeutic strategies to combat this fatal disease.

## eLife assessment

This study is **valuable** and contains results that are supported by **convincing** evidence. In the future, the observations could be further strengthened by independent validation, and by looking at larger numbers of patients, as well as by determining whether patient heterogeneity is either contributing to or obscuring certain patterns. The work will be of interest to a broad audience in the oncology and immunology fields as it is on a cancer type that does not respond well to immune checkpoint therapeutics.

## Introduction

Glioblastoma (GBM) is a fatal disease without effective long-term treatment options. The current standard of care consists of tumor resection followed by adjuvant chemoradiotherapy resulting in a median overall survival of only 14 months (*Stupp et al., 2005*). One of the hallmarks in GBM progression is the high rate of neovascularization. The GBM-induced aberrant vessels not only

nourish glioma cells, but also provide a specialized niche for tumor-associated stromal and immune cells such as monocyte-derived macrophages (MdMs), yolk sac-derived microglia (MG; together termed glioma-associated macrophages/microglia, GAMs), and peripheral adaptive immune cells. This immune tumor microenvironment (iTME) paradoxically acts in an immunosuppressive manner and promotes tumor progression (*Bowman et al., 2016*). For example, clinical trials of systemic T cell checkpoint blockade showed only disappointing results (*Reardon et al., 2017a*; *Reardon et al., 2017b*), which was attributed in part to the immunosuppressive components of the GBM iTME. The origin of GAMs, infiltration of peripherally derived macrophages across the blood-brain-barrier (BBB) or recruitment of tissue-resident MG to the tumor site, as well as their contribution to gliomagenesis are studied intensively (*Bowman et al., 2016*; *Klemm et al., 2020*; *Friebel et al., 2020*; *Müller et al., 2017*). Hence, major efforts have been undertaken to describe the GBM iTME on a single-cell level (*Klemm et al., 2020*; *Friebel et al., 2020*; *Abdelfattah et al., 2022*), or dissect the composition and changes upon disease stages, recurrence and immunotherapy specifically within the GAM compartment (*Pombo Antunes et al., 2021*; *Goswami et al., 2020*; *Chen et al., 2021*). However, human data on the composition of the iTME in different tumor regions (contrast enhancing tumor center versus peripheral infiltration zone) remain scarce (*Darmanis et al., 2017*; *Landry et al., 2020*).

To study the region-dependent cellular diversity within individual GBMs, we performed single-cell RNA sequencing (scRNA-seq) on patient-matched biopsies from the tumor center and the peripheral infiltration zone of five primary GBM patients. Additionally, peripheral blood mononuclear cells

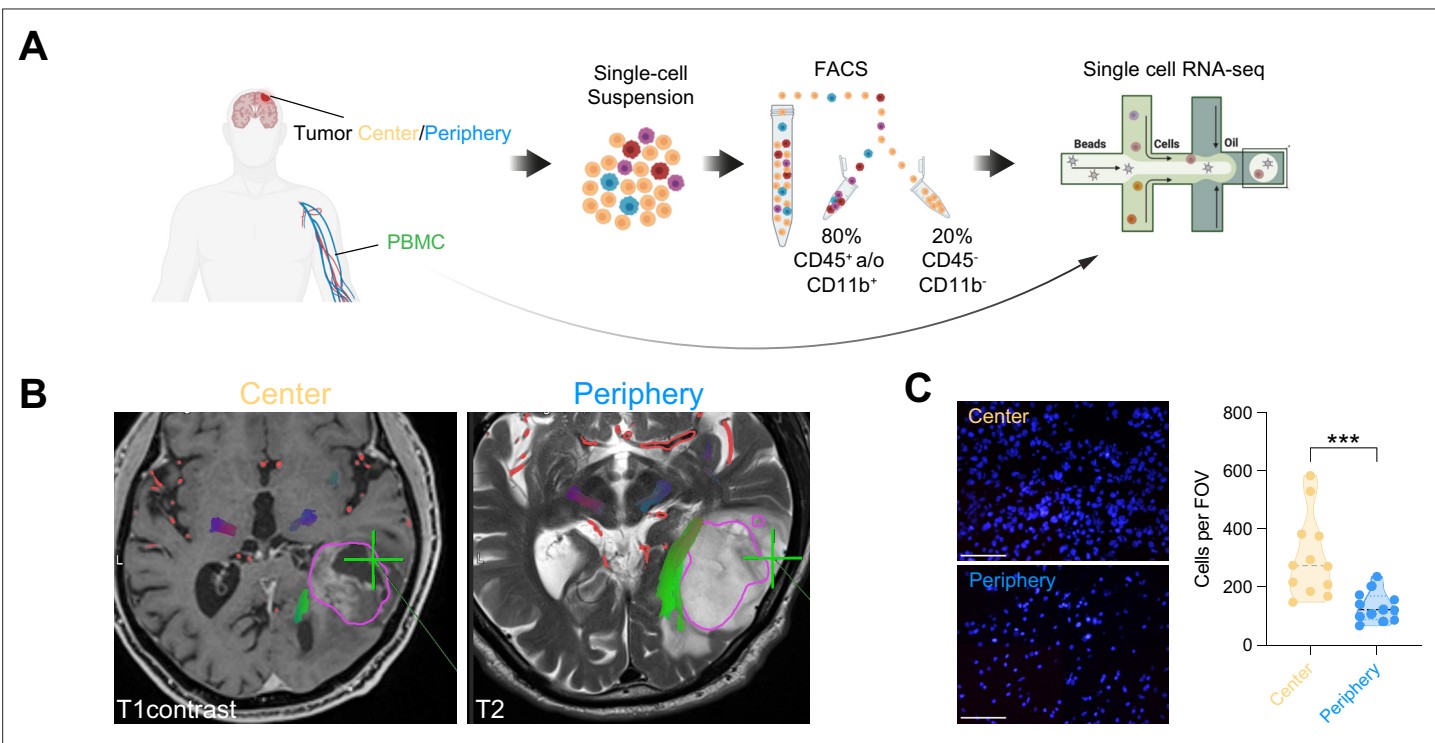

**Figure 1.** Single-cell RNA-seq of cells from tumor center, periphery and blood. (**A**) Experimental workflow for single-cell analysis of cells isolated from tumor center, periphery and peripheral blood mononuclear cells (PBMC), including fluorescent-activated cell sorting and 3'-scRNA-seq. (**B**) Axial T1 with contrast (left) and T2 (right) MRI brain in a patient with a left temporal GBM. Fresh tumor biopsies were taken according to neuronavigation (green cross). The tumor center was defined as contrast enhancing, whereas the tumor periphery was defined as T2 hyperintense. (**C**) Nuclear DAPI staining of resected tissue specimens. ×40 magnification (scale bar = 20 µm). n=3 patients, 4 field of view (FOV) per patient. Statistics: ***p<0.001, two-tailed Mann Whitney U test (*Figure 1—source data 1*).

The online version of this article includes the following source data and figure supplement(s) for figure 1:

**Source data 1.** Related to *Figure 1C*.

**Figure supplement 1.** CD45+CD11b+immune cells gating strategy and quality control of scRNA-seq data.

**Figure supplement 2.** Patient representation among clusters.

(PBMC) of the same patients were included to explore the transcriptional changes occurring during tumor infiltration of circulating immune cells.

Our analysis revealed a regionally distinct transcription profile of MG and MdMs and an impaired activation signature in the tumor-peripheral cytotoxic-cell compartment. Comparing tumor-infiltrating CD8$^+$ T cells with PBMC-derived, identified CX3CR1$^{high}$ and CX3CR1$^{int}$ CD8$^+$ T cells with effector and memory phenotype, respectively, enriched in blood but absent in the iTME. Tumor CD8$^+$ T cells displayed features of tissue-resident memory T cells and were characterized by an exhaustion phenotype. This work provides a regionally-resolved map of transcriptional states in glioma-associated cell types complemented by patient-matched PBMCs, dissecting the composition and molecular diversity of the iTME in GBM.

## Results
### scRNA-seq analysis of paired tumor center, periphery and PBMC samples

Fresh, neurosurgically resected tissue from five primary, treatment naïve GBM patients were harvested (*Figure 1A*, *Supplementary file 1*). According to the 2021 WHO Classification of Tumors of the Central Nervous System (*Louis et al., 2021*), in which the term glioblastoma designates only IDH-wildtype grade 4 tumors, we will hence use the term grade 4 glioma, as we included as well IDH-mutant grade 4 tumors (*Supplementary file 1*). The tumor center was defined as contrast enhancing, whereas the tumor periphery was defined as T2 hyperintense by magnetic resonance imaging (MRI)-guided, navigated surgical resection (*Figure 1B*). Increased cellular density of the center vs. periphery samples was confirmed by nuclear DAPI staining on matched histological micrographs of the resected tissue specimens used for scRNA-seq (*Figure 1C*). As outlined in *Figure 1A*, we separately processed patient tumor and blood samples and enriched them for immune cells by fluorescence-activated cell sorting (FACS; *Figure 1—figure supplement 1A and B*). The three samples per patient (center, periphery and PBMC) were loaded on different wells of a 10 x Genomics Chromium system for a targeted recovery of 10,000 cells. Due to technical issues cells from the center sample of patient BTB 609 could not be collected.

In total, we analyzed 45,466 cells that passed initial quality control and filtering, comprising 8254 cells from tumor center, 5954 cells from tumor periphery and 31,258 PBMCs, with 6354–10,957 cells per patient (*Supplementary file 2*; *Figure 1—figure supplement 1C–E*, *Supplementary file 3*). All cells were projected onto a two dimensions *t*-distributed stochastic neighbor embedding (tSNE; *Linderman et al., 2019*). As we observed a good overlap of cells across patients for most of the dataset (see Methods; *Figure 1—figure supplement 2B–F*), we chose not to perform any correction for patient-specific effects. Using hierarchical clustering, the cells were partitioned into clusters (*Figure 2—figure supplement 1A and B*) which were then annotated into nine distinct cell types for the immune subset, including two transcriptionally distinct MG subsets (MG_1 and MG_2) and four cell types for the CD45-negative subset (*Figure 2A*; *Figure 2—figure supplement 1C and D*; *Supplementary file 3*).

In more detail, our annotation strategy made use of the relative expression patterns of known marker genes (*Figure 2B*), and of cluster-specific genes (*Figure 2—figure supplement 2*). Additionally, unbiased cell-type annotation using whole-transcriptome comparisons to reference bulk and single-cell datasets was performed with the *SingleR* package. We used as reference (i) a public bulk RNA-seq dataset of sorted immune cell types from human PBMC samples (*Monaco et al., 2019*), which helped in identifying the major immune cell lineages: B cells, CD4$^+$ and CD8$^+$ T cells, dendritic cells (DCs), monocytes and NK cells (*Figure 2—figure supplement 1E*). (ii) A bulk RNA-seq dataset of sorted immune cell types from the TME of human gliomas (*Klemm et al., 2020*) was helpful to separate GBM-associated CD4$^+$ and CD8$^+$ T cells, as well as microglia and MdMs (*Figure 2—figure supplement 1F*). (iii) A 10 X genomics scRNA-seq dataset of the innate immune TME of seven newly diagnosed GBM patients (*Pombo Antunes et al., 2021*) confirmed the annotation of the innate immune subset (*Figure 2—figure supplement 1I*), which was also supported by a microglia and a macrophage signature scores defined using gene lists obtained from *Müller et al., 2017*; *Figure 2—figure supplement 1J and K*. (v) Finally, a Smartseq2 scRNA-seq dataset of *IDH1$^{wt}$* tumors (*Neftel et al., 2019*) was used to characterize the CD45$^{neg}$ population (*Figure 2—figure supplement 1N*).

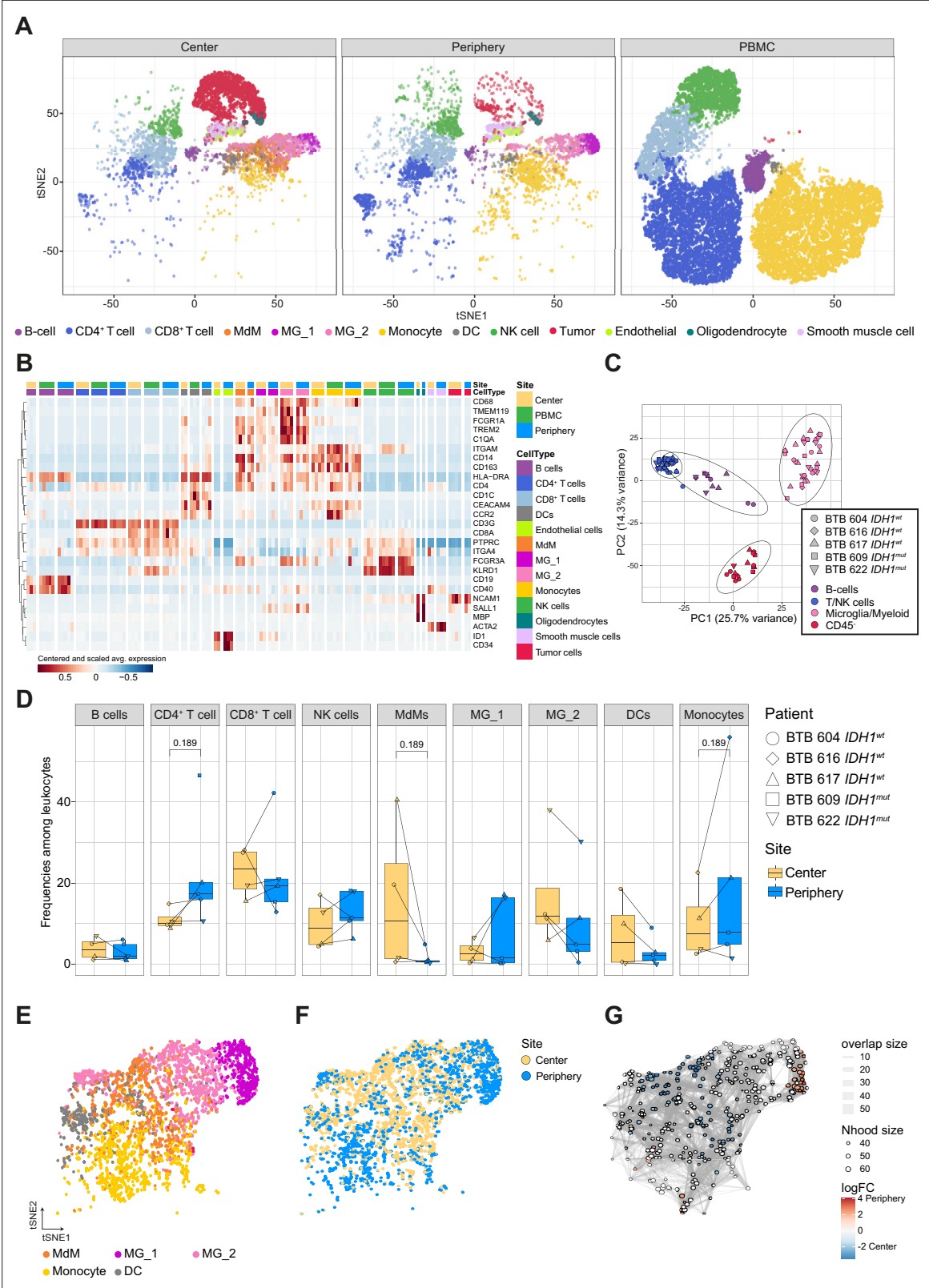

**Figure 2.** Single-cell RNA-seq analysis identifies main immune cell populations. (**A**) Dimensionally reduced tSNE projection of the scRNAseq data showing the annotated cell types. (**B**) Heatmap displaying centered and scaled normalized average expression values of characteristic cell-type specific genes used to annotate clusters. Columns are ordered by site and cell type, and rows show centered and scaled expression values, hierarchically clustered. Heatmap displaying genes whose expression is most specific to each cell type is shown in *Figure 2—figure supplement 2*. (**C**) Principal

*Figure 2 continued on next page*

*Figure 2 continued*

component analysis of pseudo-bulk scRNAseq samples aggregated by patient and cell type. Symbols represent individual patients and cell lineage is displayed by different colors. (**D**) Relative frequencies of immune populations among leukocytes between tumor center and periphery, shown as boxplots. Symbols represent individual patients (n=5) and paired samples are indicated by connecting lines. p-Values were calculated using *diffcyt-DA-voom* method (*Figure 2—source data 1*). (**E–G**) Differential abundance testing of the tumor innate immune compartment (**E**) using the *miloR* package which tests the abundance of each neighborhood of cells separately between tumor center und periphery (**F, G**).

The online version of this article includes the following source data and figure supplement(s) for figure 2:

**Source data 1.** Related to *Figure 2D*.

**Figure supplement 1.** Cross-referencing scRNA-seq data with published datasets.

**Figure supplement 2.** Cell type specific gene expression.

To perform a differential expression analysis between tumor sites, we stratified the analysis by cell type to decrease the influence of differential abundance patterns (analyzed separately, see below). A principal component analysis (PCA) on the pseudo-bulk aggregated samples (see Methods) confirmed that the major source of variation was due to differences between lymphoid, MG/myeloid cells, and CD45$^-$ cells (PCs 1 and 2; *Figure 2C*). There was no clear association to the *IDH1* status of patients on these or deeper components. However, our study was neither designed nor powered to find regional signatures within the iTME depending on *IDH1* status, but rather to identify common transcriptional differences within the iTME between tumor center, periphery and PBMC of *IDH1$^{wt}$* and *IDH1$^{mut}$* grade 4 glioma.

Differential abundance analysis between tumor center and periphery did not reveal significant changes, but some populations displayed suggestive differences: the abundance of both CD4$^+$ T cell and monocytic clusters increased in periphery compared to center, while the abundance of MdM clusters decreased (*Figure 2D*). This has been observed by others as well (*Darmanis et al., 2017*; *Pinton et al., 2019*). Testing the abundance at a finer scale with *miloR* provided us interesting insights. Indeed, within the innate immune population (*Figure 2E*) we confirmed the decreased abundance of MdMs in the tumor periphery and noticed an increased abundance of the MG_1 subset of MG in the tumor periphery. (*Figure 2F and G*).

## MG display regionally resolved transcriptional profiles that differ from those of DCs and MdMs

Differential expression analysis between MG from tumor center and periphery revealed downregulation of inflammatory-related genes in the peripheral MG (*FCGBP* and *CCL20*), downregulation of genes associated with canonical interferon (IFN) responses (*IFI6, IFI27, STAT1, ISG15*), and cell proliferation (*STMN1*) as well as downregulation of scavenger receptor gene *CD163* (*Figure 3A* and *Supplementary file 4*). The latter was shown to have a potential role in the phagocytic response of MG to beta-amyloid depositions identified in single-nucleus RNA-seq of postmortem human brain (*Nguyen et al., 2020*).

Among many upregulated genes whose function is not yet known, we found Inhibitor of DNA-Binding 1, also known as Inhibitor of Differentiation 1 (*ID1*) to be increased in the peripheral MG (*Figure 3A*). ID1 is well described in GBM progression, treatment resistance and glioma stem cell biology (*Soroceanu et al., 2013*). However, new evidence has emerged, linking *ID1* to suppression of the anti-tumor immune response in the myeloid compartment and promoting tumor progression (*Papaspyridonos et al., 2015*).

To further explore the underlying biological processes differing between MG in the two compartments, we conducted a gene set enrichment analysis (GSEA) on the results of the differential expression analysis using Gene Ontology (GO) database (Biological Processes). This revealed overall a significant downregulation of GO categories involved in antigen processing and presentation via MHC-I and MHC-II in the peripheral MG relative to the center MG, as well as downregulation of amino acid metabolism and TNF-α signaling pathway (*Figure 3B*), which supported the conclusion of an impaired activation state of peripheral MG.

When comparing the transcriptional profiles of the other innate immune phagocytic populations, DCs and MdMs, we observed for both cell types upregulation of pro-inflammatory related gene sets in the tumor periphery compared to tumor center (*Figure 3C and D, Figure 3—figure supplement*

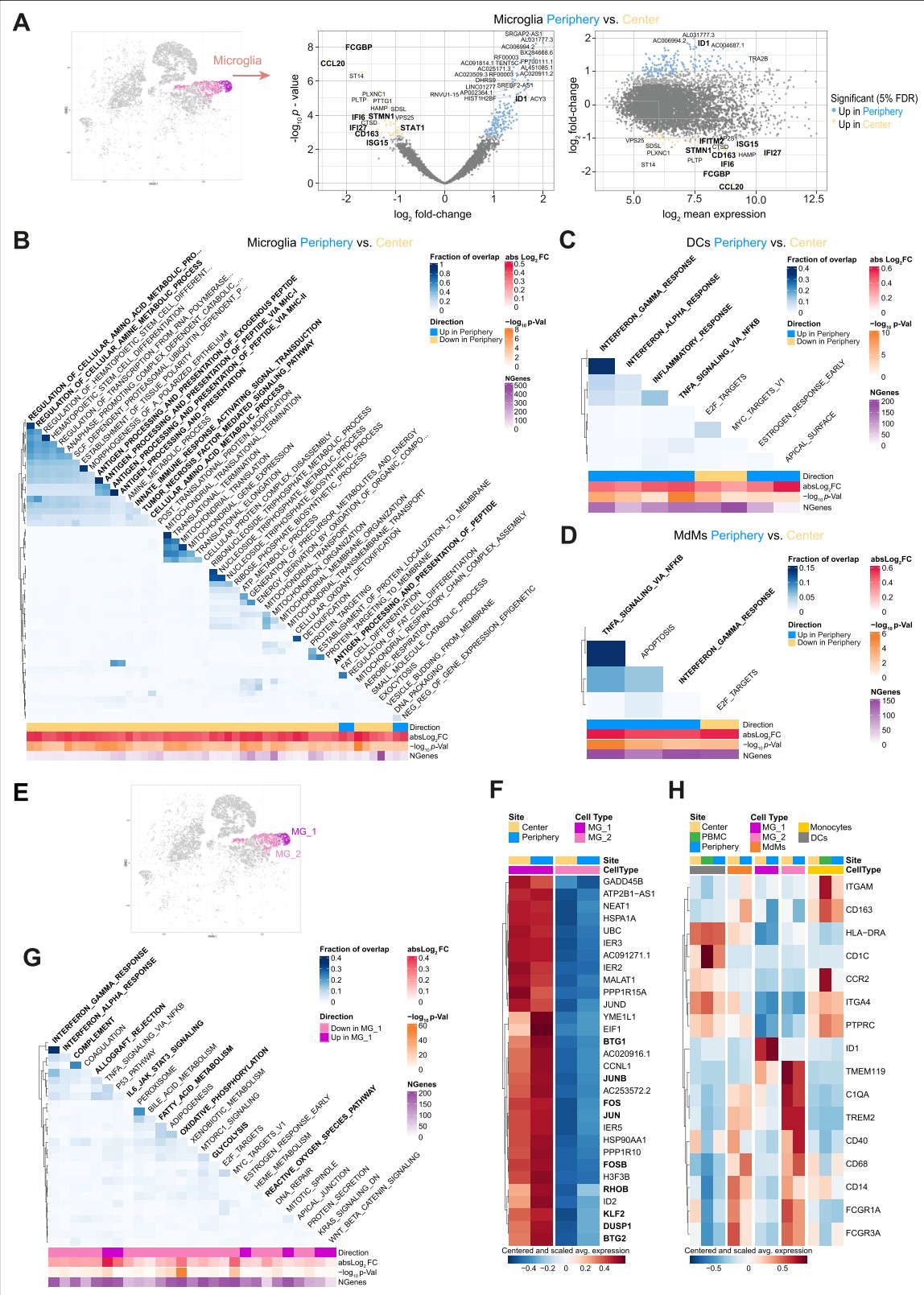

**Figure 3.** MG display regionally resolved transcriptional profiles that differ from those of DCs and MdMs. (**A**) Microglia cluster highlighted on tSNE map and scatterplots showing differentially expressed genes (FDR <5%, indicated by blue and yellow) in Microglia (MG) cells from tumor periphery versus center. Volcano plot showing *p* value versus fold-change (left) and MA plot showing fold-change versus mean expression (right). For a complete list of differentially expressed genes per cell cluster between tumor periphery and center, please refer to ***Supplementary file 4***. (**B**) Heatmap representation

*Figure 3 continued on next page*

Figure 3 continued

of Gene set enrichment analysis (GSEA) results between peripheral and center microglia using Gene Ontology (GO) collection (Biological Processes). The fraction of overlap between gene sets is calculated as Jaccard coefficient of overlap between the gene sets. (C, D) Heatmap representation of GSEA of DCs (C) and monocyte-derived macrophages (MdMs) (D) from tumor periphery versus tumor center using Hallmark collection of major biological categories. (E) Unsupervised hierarchical sub-clustering of the MG population revealed two transcriptionally distinct subsets of MG, termed MG_1 and MG_2, displayed on the tSNE map. (F) Heatmap displaying the cluster-specific genes identifying MG_1 subcluster. Columns are ordered by site and cell type, and rows show centered and scaled normalized average expression values, hierarchically clustered. A complete list of cluster specific genes for MG_1 and MG_2 subcluster is provided in *Supplementary file 5*. (G) Heatmap representation of GSEA between MG_1 and MG_2 subclusters using Hallmark collection of major biological categories. (H) Heatmap displaying previously described reactivity markers of MG. Columns are ordered by site and cell type, and rows show centered and scaled normalized average expression values, hierarchically clustered.

The online version of this article includes the following figure supplement(s) for figure 3:

**Figure supplement 1.** Regionally dependent transcriptional profiles of innate immune subsets and MG subclusters MG_1 and MG_2.

*1A and B* and *Supplementary file 4*). Within DCs, the induced pro-inflammatory phenotype was also observed when comparing DCs from tumor periphery to PBMC-derived ones (*Figure 3—figure supplement 1C and D* and *Supplementary file 6*). Taken together, DCs and MdMs seem to have a proinflammatory phenotype in the glioma periphery as opposed to MG. However, they are less abundant there, at least as far as MdMs are concerned (*Figure 2E–G*).

## The iTME of grade 4 glioma harbors two transcriptionally distinct MG subpopulations

Unsupervised hierarchical sub-clustering of the MG population revealed two transcriptionally distinct iTME MG subsets, which we termed MG_1 and MG_2, respectively (*Figure 3E*, *Figure 3—figure supplement 1E* and *Supplementary file 5*). The MG_1 cluster was highly enriched for the activator protein-1 (AP-1) family of transcription factors including *FOS, FOSB, JUN, JUNB, MAF,* and *MAFB* (*Figure 3F* and *Supplementary file 5*), which convey a surveilling phenotype to adult MG, but are also involved in numerous processes including cell growth, differentiation, and immune activation (*Holtman et al., 2017*). Specifically, *FOSB* gene products were implicated in the excitotoxic MG activation by regulating complement C5a receptor expression (*Nomaru et al., 2014*). Yet, concomitant upregulation of anti-inflammatory Krüppel-like factor 2 (*KLF2*) (*Sweet et al., 2020*) and Dual Specificity Protein Phosphatase 1 (*DUSP1*), an inhibitor of innate inflammation by negatively regulating the mitogen-activated protein kinase (MAPK) pathway (*Salojin et al., 2006*), together with increased expression of anti-proliferative genes like *RHOB, BTG1* and *BTG2* paint a more complex picture of these cells. Particularly, *BTG1* was identified as an activation-induced apoptotic sensitizer in MG after exposure to inflammatory stimuli (*Lee et al., 2003*), serving as an autoregulatory mechanism and possibly hinting towards an exhausted state in these MG_1 cells. GSEA for differences between MG_1 and MG_2 clusters using the MSigDB Hallmark collection of major biological pathways (*Liberzon et al., 2015*) revealed downregulation of many MG effector functions in the MG_1 population including (1) inflammation ('Complement', 'Allograft Rejection', 'Reactive Oxygen Species Pathway'), (2) immune cell activation ('IFN-α Response', 'IFN-γ Response', 'IL6 JAK STAT3 Signaling'), and (3) immunometabolism ('Fatty Acid Metabolism', 'Oxidative Phosphorylation', 'Glycolysis'; *Figure 3G*). As we examined the expression of previously described reactivity markers of MG including *C1QA, FCGR1A, CD14, HLA-DRA, TREM2,* and Ferritin (*FTH1*) (*Walker and Lue, 2015*; *Hopperton et al., 2018*; *Hammond et al., 2019*; *McQuade et al., 2020*; *Figure 3H* and *Figure 3—figure supplement 1F*), and established MG homeostatic genes like *CX3CR1, HEXB* and *SPI1* (PU.1) (*Supplementary file 5*), we noted a reduced expression of these genes in the MG_1 cluster compared to MG_2 cells, while the anti-inflammatory transcription factors *NR4A1* (*Rothe et al., 2017*) *NR4A2* (*Saijo et al., 2009*) were highly upregulated (*Figure 3—figure supplement 1G*). Additionally, when comparing the distribution of MG subsets between sites among total MG, we observed an increased abundance of MG_1 cells in the tumor periphery, which might at least partially explain the previously observed impaired activation state of total peripheral MG (*Figure 3—figure supplement 1H*). To rule out this explanation, we stratified the differential expression analysis between tumor periphery and center by MG subtype and observed similar genes downregulated in the tumor periphery in each subtype (e.g. *CCL20, FCGBP*; *Supplementary file 5*), arguing that the non-reactive phenotype is a common feature

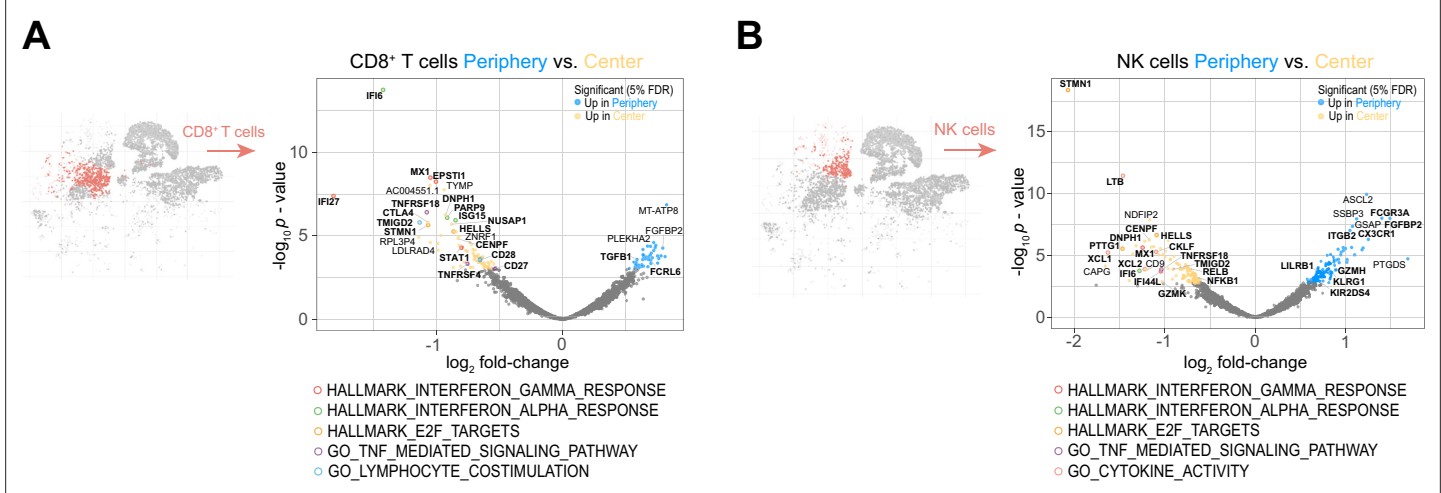

**Figure 4.** The peripheral cytotoxic cell compartment exhibits an impaired activation signature. (**A, B**) Volcano plots showing differentially expressed genes (FDR corrected p-value <0.05, indicated by blue and yellow) in CD8+ T cells (**A**) and NK cells (**B**) from tumor periphery versus tumor center. Colored rings mark genes belonging to selected GSEA Hallmark or Gene Ontology (GO) pathways as indicated. For a complete list of differentially expressed genes per cell cluster between tumor periphery and center, please refer to *Supplementary file 4*.

The online version of this article includes the following figure supplement(s) for figure 4:

**Figure supplement 1.** Differential expression analysis between tumor center and peripheral CD4+T cells.

## The tumor peripheral cytotoxic cell compartment exhibits an impaired activation signature

Next, we investigated the regional differences in the lymphoid compartment composed of CD4+ and CD8+ T cells and natural killer (NK) cells. We observed only very few significant changes in the transcriptomic profiles of CD4+ T cells between tumor center and periphery (*Figure 4—figure supplement 1A* and *Supplementary file 4*). Yet, comparing peripheral CD8+ T cells with CD8+ T cells from tumor center revealed 110 differentially expressed genes (43 genes upregulated and 67 genes downregulated; *Figure 4A* and *Supplementary file 4*). Many downregulated genes in the peripheral CD8+ T cells associated with canonical IFN responses (*IFI6, IFI27, MX1, STAT1, EPSTI1 PARP9, ISG15*; *Szabo et al., 2019*), cell proliferation (*STMN1, CENPF, HELLS, NUSAP1,* and *DNPH1*) and T cell co-stimulation (*CD28, TMIGD2* [CD28H], *TNFRSF4* [OX40], *CD27* and *TNFRSF18* [GITR]; *Figure 4A*). Contrary to our expectations, we saw upregulation of *CTLA4* in the center CD8+ T cells which acts as a negative costimulatory molecule. However, unlike other costimulatory receptors, such as CD27 and CD28, CTLA-4 is not constitutively expressed on T lymphocytes (*Alegre et al., 1996*). but only induced following T cell activation, along with positive costimulatory molecules such as OX40 and GITR. In addition, upregulation of CTLA-4 requires entry into the cell cycle (*Alegre et al., 1996*). In line with that, we detected an upregulation of proliferative genes in center CD8+ T cells. In summary, CTLA-4 induction in center CD8+ T cells rather suggested T cell activation than exhaustion, especially since other inhibitory receptors like *PDCD1* (PD-1), *LAG3* and *HAVCR2* (TIM-3) were not differentially expressed between sites. Moreover, we did not observe differential expression of genes involved in CD8+ T cell effector functions like cytotoxicity (e.g. *GZMK, GZMB, KLRG1,* PRF1) or cytokines (e.g. *CCL5, XCL1, XCL2, IL10*). Yet, we noted upregulation of inhibitory genes (*TGFB1* and *FCRL6 Johnson et al., 2018*) in the peripheral CD8+ T cells, suggesting that a pool of activated, proliferating and IFN-responsive CD8+ T cells is present in the tumor center, however, absent in the tumor periphery.

Similar trends were observed for the peripheral NK cell population with peripherally reduced IFN response (*MX1* and *IFI44L*), and proliferative genes (*STMN1, HELLS, CENPF, PTTG1,* and *DNPH1*), downregulated stimulatory receptors (*TMIGD2* [CD28H] and *TNFRSF18* [GITR]), and reduced NF-κB signaling (*NFKB1* and *RELB*; *Figure 4B* and *Supplementary file 4*). Although, we observed upregulation

of key genes associated with NK cell effector function in the periphery (e.g. *FCGR3A* [CD16], *FGFBP2*, *ITGB2*, *GZMH,* and *KIR2DS4*), increased expression of inhibitory receptors like *LILRB1* and *KLRG1*, the latter especially in co-expression with chemokine receptor *CX3CR1*, identified the peripheral NK cells rather to be terminally differentiated with impaired cytotoxic capabilities (*Sciumè et al., 2011*). This was in line with the observed abrogated cytokine activity profile in the peripheral NK cells with reduced expression of key factors like *XCL1, XCL2, LTB,* and *CKLF*. In summary, our data revealed an impaired activation signature in the peripheral cytotoxic cell compartment.

## CX3CR1 labels a specific CD8[+] T cell population in the circulation of grade 4 glioma patients

Next, we investigated the relationships between circulating CD8[+] T cells and those from the tumor milieu and, more specifically, the peripheral, infiltration zone characterized by an abrogated CD8[+] T-cell IFN response and activation signature. Strikingly, there were large transcriptomic differences between PBMC and periphery CD8[+] T cells (*Figure 5A*), with 1,417 differentially expressed genes (864 genes upregulated in the tumor periphery and 553 genes upregulated in PBMC; *Figure 5B*, *Supplementary file 6*).

Interestingly, one of the key genes upregulated in PBMC CD8[+] T cells was the chemokine receptor *CX3CR1* (*Figure 5B*). Flow cytometry of an additional matched glioma grade 4 patient cohort confirmed an increased abundance of CX3CR1[+] CD8[+] T cells in PBMC compared to almost absent CX3CR1[+] CD8[+] T cells in tumor periphery (*Figure 5C*, *Figure 5—figure supplement 1A*, *Supplementary file 1*). Unsupervised hierarchical sub-clustering of CD8[+] T cells revealed that *CX3CR1* expression labelled a specific subpopulation among PBMC CD8[+] T cells (*Figure 5D and E*).

Recently, expression of CX3CR1 was demonstrated to distinguish memory CD8[+] T cells with cytotoxic effector function in healthy donors and different inflammation related conditions (*Böttcher et al., 2015*; *Gerlach et al., 2016*; *Yamauchi et al., 2021*; *Yan et al., 2018*). Further characterization of classical central memory (T$_{cm}$) and effector memory (T$_{em}$) populations by varying surface expression levels of CX3CR1 identified a novel CX3CR1$^{int}$ subpopulation, termed peripheral memory (T$_{pm}$). T$_{pm}$ cells underwent frequent homeostatic divisions, re-acquired CD62L, homed to lymph nodes, and predominantly surveyed peripheral tissues compared to T$_{cm}$ and T$_{em}$ (*Gerlach et al., 2016*). In our dataset, the circulating CX3CR1[+] CD8[+] T cells indeed displayed a core signature of memory CD8[+] T cells with effector function, comprising expression of LFA-1 (*IGAL- ITGB2*), *TBX2* (Tbet), *SELL* (CD62L), *GZMB* (Granzyme B), and *PRF1* (Perforin 1; *Figure 5F*), separating them from circulating *CX3CR1[-] CD28$^{high}$*, *IL7R$^{high}$* and *CD27$^{high}$* naïve CD8[+] T cells (*Figure 5G* and *Figure 5—figure supplement 1B*). Flow cytometric analysis confirmed T$_{eff}$ to be CX3CR1$^{high}$, with negligible expression levels in the naïve CD8[+] T cells, whereas the identified memory CD8[+] T cells (T$_{em}$ and T$_{pm}$) were CX3CR1$^{int}$ (*Figure 5H and I*). Collectively, surface expression analysis of CX3CR1 identified a subset of CX3CR1$^{high}$ T$_{eff}$ and CX3CR1$^{int}$ memory (T$_{em}$, T$_{pm}$) CD8[+] T cells in the circulation of grade 4 glioma patients with potentially elevated tissue surveilling properties in the case of T$_{pm}$, which are, however, largely absent in the tumor microenvironment.

## CD8[+] T cells in the tumor periphery share features with tissue-resident memory T cells (T$_{rm}$)

We next examined the differing transcriptional and surface-specific features between tumor infiltrating and circulating CD8[+] T cells. Surface staining for CD45RA and CD45RO, discriminating naïve/effector from memory T cells, attributed a predominant CD45RO[+] memory phenotype to the tumor infiltrating CD8[+] T cells (*Figure 6A and B*). Interrogation of the transcriptomic profile of these cells revealed a key marker expression signature consistent with tissue-resident memory T cells (T$_{rm}$): Expression of cellular adhesion molecules (integrins) *ITGA1* (CD49a) and *ITGAE* (CD103), tissue retention marker *CD69*, chemokine receptors implicated in tissue-homing *CXCR3, CXCR6,* and *CCR5* (*Urban et al., 2020*) and transcription factors, *ZNF683* (Hobit) and *PRDM1* (Blimp1) as well as reduced expression of *TBX21* (Tbet) and *EOMES* (*Mackay et al., 2016*), strongly suggested a T$_{rm}$ phenotype for these cells (*Figure 6C and D* and *Figure 6—figure supplement 1A*). Co-expression analysis of paired PBMC and tumor periphery samples using flow cytometry showed that CD69[+] CD103[-] and CD69[+] CD103[+] cells are the dominant CD8[+] T cell populations in the tumor periphery (*Figure 6E and F*). Combined, these data strongly suggest a T$_{rm}$ phenotype for the CD8[+] T cells in the tumor periphery.

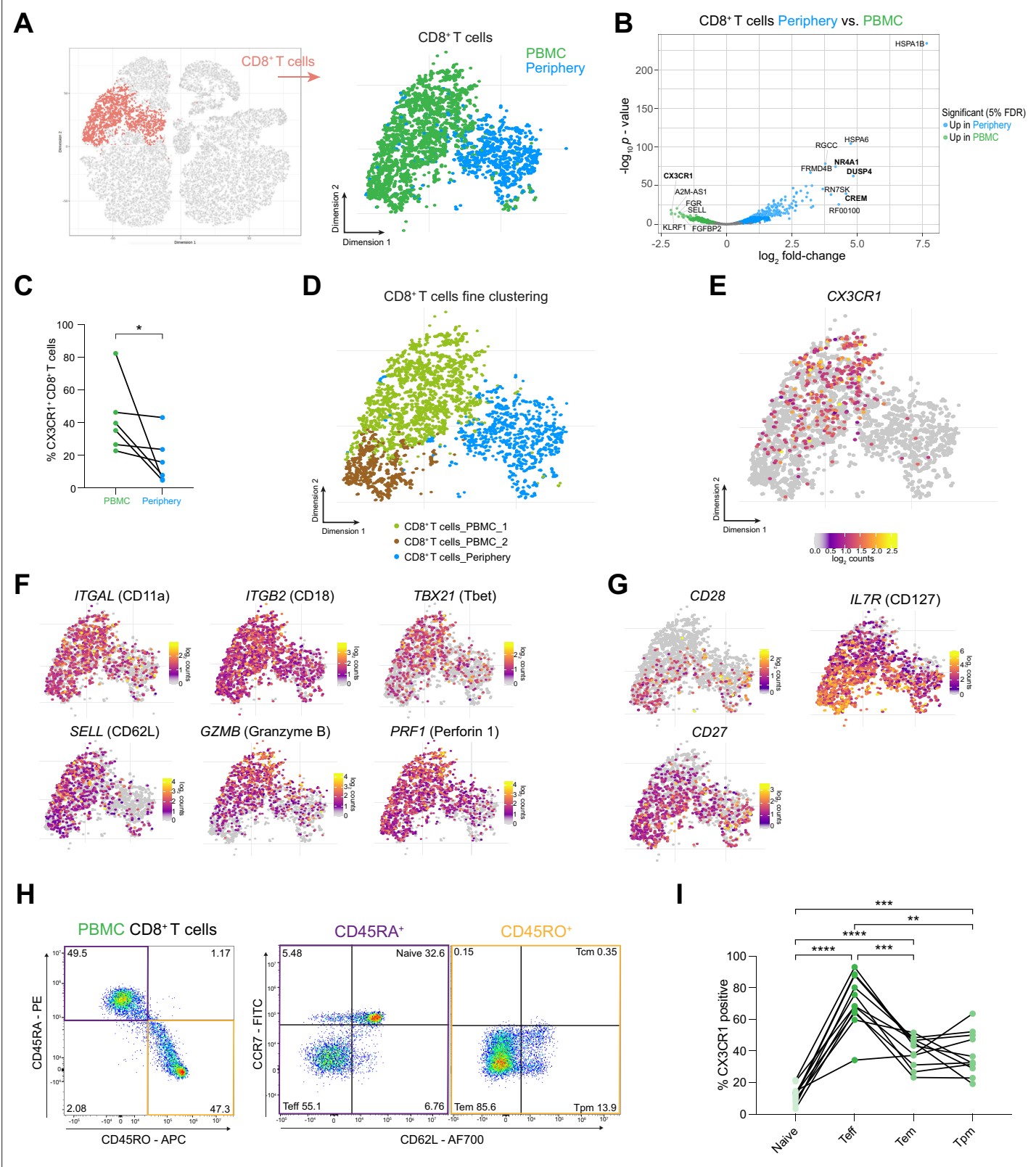

**Figure 5.** CX3CR1 labels a specific CD8 +T cell population in the circulation of grade 4 glioma patients. (**A**) CD8+ T cell cluster highlighted on tSNE map (left). CD8+ T cell cluster colored by site of origin (right). (**B**) Volcano plot showing differentially expressed genes (FDR corrected p-value <0.05, indicated by blue and green) in CD8+ T cells from tumor-periphery versus PBMC. For a complete list of differentially expressed genes per cell cluster between tumor periphery and PBMC, please refer to ***Supplementary file 6***. (**C**) Frequency of CX3CR1+ CD8+ T cells among all CD8+ T cells in flow

*Figure 5 continued on next page*

*Figure 5 continued*

cytometry data (*Figure 5—source data 1*). (**D**) Unsupervised hierarchical sub-clustering of CD8⁺ T cells from PBMC and Periphery revealed two transcriptionally distinct subsets of PBMC CD8⁺ T cells, displayed on the tSNE map. (**E**) Expression of *CX3CR1* overlaid on tSNE CD8⁺ T cell cluster. (**F**) Expression of genes associated with effector memory phenotype overlaid on tSNE CD8⁺ T cell cluster. Displayed genes are significantly, differentially expressed genes (DEGs) between tumor periphery and PBMC, as identified by differential gene expression analysis shown in panel (**B**). (**G**) Expression of selected genes associated with naive phenotype overlaid on tSNE CD8⁺ T cell cluster (**H**) Gating procedure applied to identify CD3⁺ CD8⁺ naive, T effector cells ($T_{eff}$), effector memory ($T_{em}$), peripheral memory ($T_{pm}$) and central memory ($T_{cm}$), eluted from PBMCs. (**I**) Expression of CX3CR1 in PBMC CD8⁺ T cell subpopulations identified in (**H**) (*Figure 5—source data 2*). n=6 donors (**C**), n=11 donors (**I**). Statistics: Wilcoxon matched-pairs signed rank test (**C**); repeated measures one-way ANOVA with post-hoc Šidák's correction for multiple comparisons (**I**). *p≤0.05, **p≤0.01, ***p≤0.001, ****p≤0.0001, no brackets indicate no significant difference.

The online version of this article includes the following source data and figure supplement(s) for figure 5:

**Source data 1.** Related to *Figure 5C*.

**Source data 2.** Related to *Figure 5I*.

**Figure supplement 1.** Phenotypic characterization of PBMC CD8⁺ T cells.

Previous reports of $T_{rm}$ populating the brain in the aftermath of central or peripheral infections concluded that brain $T_{rm}$ cells surveil the brain tissue and mediate protection by rapid activation and enhanced cytokine production (*Urban et al., 2020*). Indeed, CD8⁺ T cells in the tumor periphery showed increased expression of genes belonging to costimulatory pathways, including *ICOS*, *TNFRSF4* (OX40) and *TNFRSF9* (4-1BB) (*Figure 6—figure supplement 1B*, *Supplementary file 6*), albeit accompanied by high levels of inhibitory receptors *PDCD1* (PD-1), *LAG3*, *HAVCR2* (TIM-3), and *CTLA4* (*Figure 6G*). Moreover, expression of genes coding for cytotoxic molecules, including Granzyme B and Perforin 1 were decreased in the peripheral CD8⁺ T cells, suggesting a compromised killing capacity of these cells. And lastly, CD8⁺ T cells in the tumor periphery exhibited a transcription factor profile of exhausted T cells with high expression of *NR4A1*, *MAF*, and *IRF4* (*Figure 6G* and *Figure 6—figure supplement 1C*), which were implicated in T cell dysfunction and exhaustion (*Ma et al., 2019*; *Liu et al., 2019*). Collectively, these data indicate that CD8⁺ T cells in the glioma periphery share features with $T_{rm}$ cells. However, inhibitory receptor expression, functional molecules and transcriptional signature ascribe an exhausted phenotype to these cells.

Noteworthy, we observed high upregulation of similar genes in the comparison tumor periphery vs. PBMC for CD4⁺ T cells as for CD8⁺ T cells (*Figure 6—figure supplement 1D*). These included transcription factor family *NRA41-3*, identified as key mediator of T cell dysfunction (*Liu et al., 2019*), Dual Specificity Protein Phosphatase 2/4 (*DUSP2*, *DUSP4*) described as negative regulators of mitogen-activated protein (MAP) kinase superfamily and associated with impaired T cell effector activity (*Dan et al., 2020*) and T cell senescence (*Bignon et al., 2015*), and transcription factor *CREM* which was implicated in IL-2 suppression (*Maine et al., 2016*). These genes could potentially identify pan T cell dysfunction markers within the GBM iTME (*Li et al., 2019*).

## Interrogation of cell-cell interactions revealed critical role of SPP1-mediated crosstalk between MG and lymphocytes in the tumor periphery

We finally investigated cell-cell interactions based on ligand-receptor expression levels using the Cell-Chat platform (*Jin et al., 2021*). Considering that MG and lymphocytes displayed an impaired activation signature in the tumor periphery, we focused our analysis on the tumor-peripheral crosstalk between these cells (*Figure 7A*). This revealed SPP1 (Osteopontin) as a leading potential cell-cell interaction mediator between MG and lymphocytes (*Figure 7A and B*). MG SPP1-mediated signaling was as well among the most significant interactions, when investigating cell-cell communication across all cell types and both sites (*Figure 7—figure supplement 1A and B*). Further, we found that *SPP1* is mainly expressed by MG rather than glioma cells, contrary to previous reports (*Wei et al., 2019*; *Figure 7C*, *Figure 7—figure supplement 1C and D*). MG SPP1 conveys different interactions, depending on the recipient cell binding receptor expression profile. The predicted interactions of MG SPP1 with NK cells could be mediated via the integrin complex *ITGA4-ITGB1* (CD49d-CD29) (*Figure 7A and C*), whereas CD4⁺ and CD8⁺ T cells exhibited strong interactions with MG via SPP1/ CD44 interaction (*Figure 7A and B*). The SPP1/CD44 axis was recently described to suppress T cell

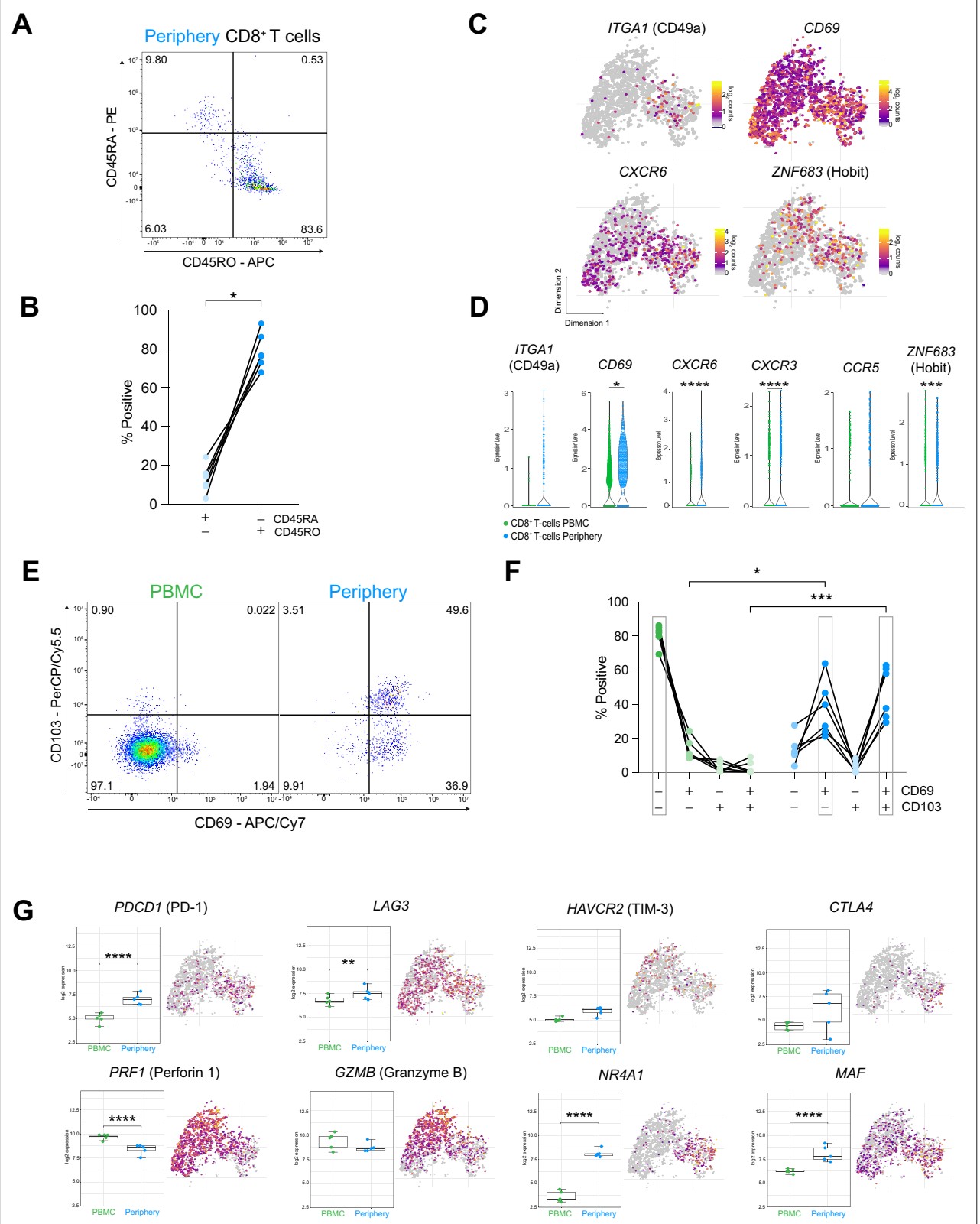

**Figure 6.** CD8[+] T cells in the tumor periphery share features with tissue-resident memory T cells (Trm). (**A**) Representative dot plot of tumor-periphery CD8[+] T cells stained for CD45RA and CD45RO. (**B**) Quantification of tumor-periphery CD8[+] T cells expressing CD45RA or CD45RO (***Figure 6—source data 1***). (**C**) Expression of genes associated with tissue-resident memory (T$_{rm}$) phenotype overlaid on tSNE CD8[+] T cell cluster. (**D**) Average expression levels of selected T$_{rm}$ markers between CD8[+] T cells from PBMC versus tumor-periphery. Significance testing based on differential gene expression

*Figure 6 continued on next page*

*Figure 6 continued*

analysis shown in panel (*Figure 5B*) (**E**) Representative dot plots of CD69 and CD103 co-expression in CD8+ T cells from PBMC and tumor-periphery. (**F**) Quantification of CD69 and CD103 co-expression revealed CD69- CD103- in PBMC and CD69+ CD103- and CD69+ CD103+ in tumor-periphery as the dominant phenotypes (*Figure 6—source data 2*). (**G**) Expression of selected markers associated with T cell exhaustion/dysfunction, shown as boxplots between CD8+ T cell from PBMC and tumor-periphery and overlaid on tSNE CD8+ T cell cluster. Significance testing based on differential gene expression analysis shown in panel (*Figure 5B*). n=6 donors (**B, F**). Statistics: Wilcoxon matched-pairs signed rank test (**B**); repeated measures one-way ANOVA with post-hoc Šidák's correction for multiple comparisons (**F**). *p≤0.05, **p≤0.01, ***p≤0.001, ****p≤0.0001, no brackets indicate no significant difference.

The online version of this article includes the following source data and figure supplement(s) for figure 6:

**Source data 1.** Related to *Figure 6B*.

**Source data 2.** Related to *Figure 6F*.

**Figure supplement 1.** Phenotypic characterization of Periphery CD8+ T cells.

activation and proliferation (*Klement et al., 2018*). The identified MG SPP1-mediated interactions might represent potential targets to modulate MG-lymphocyte crosstalk in the tumor periphery.

## Discussion

In this study, we combined single-cell RNA sequencing and flow cytometry to interrogate the regional leukocyte activation signature in patient-matched biopsies from contrast-enhancing tumor center, infiltrative peripheral rim, and blood PBMCs of grade 4 glioma patients. Our analyses revealed a distinct, regionally dependent transcriptional profile for most of the investigated cell populations. While peripheral MG and cytotoxic cells predominantly displayed an impaired activation signature, MdMs showed pro-inflammatory traits in the tumor periphery, however, were less abundant there compared to the tumor center, which was reported by others as well (*Darmanis et al., 2017*; *Landry*

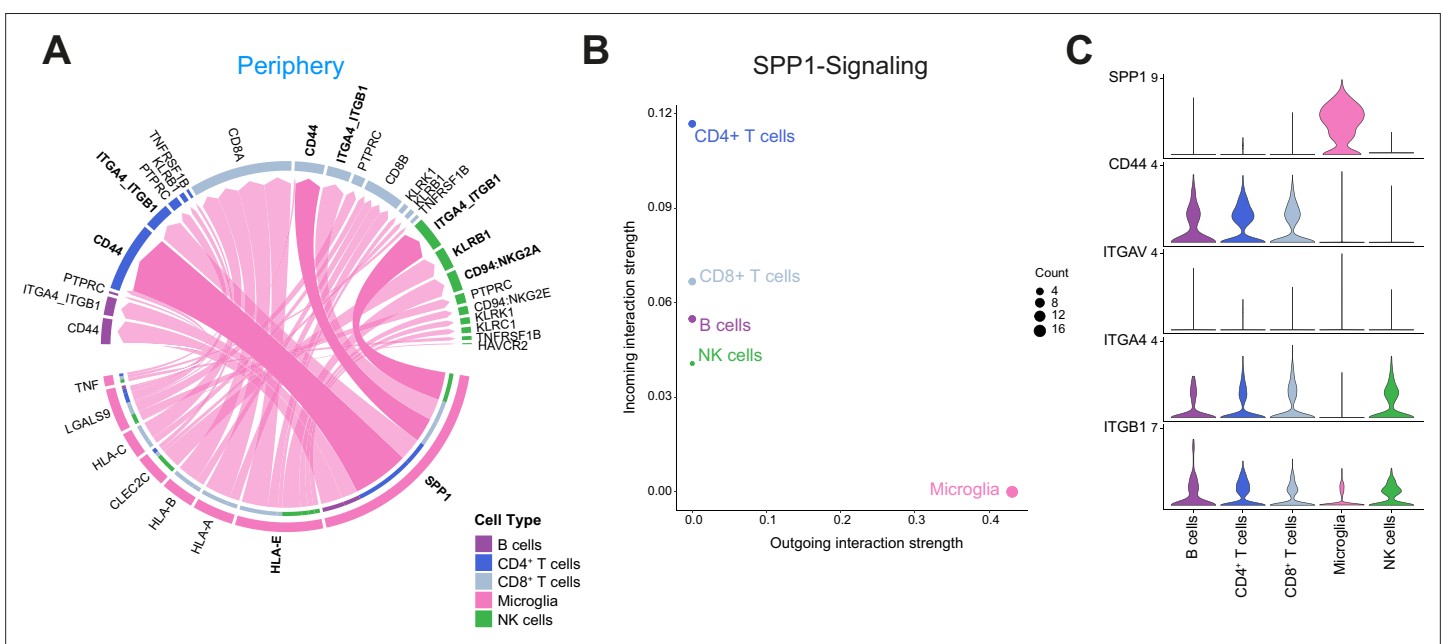

**Figure 7.** Cell-cell communication analysis using CellChat reveals critical role for SPP1-mediated crosstalk in tumor periphery. (**A**) Chord diagram showing significant interactions from microglia to lymphocyte cell clusters. The inner bar colors represent the targets that receive signal from the corresponding outer bar. The inner bar size is proportional to the signal strength received by the targets. Chords indicate ligand-receptor pairs mediating interaction between two cell clusters, size of chords is proportional to signal strength of the given ligand-receptor pair. (**B**) Comparison of incoming and outgoing interaction strength allows identification of main senders and receivers. (**C**) Violin plots showing the expression distribution of signaling genes involved in the inferred SPP1 signaling network.

The online version of this article includes the following figure supplement(s) for figure 7:

**Figure supplement 1.** Cell-cell communication analysis using CellChat.

*et al., 2020*). Supplemented with transcriptional analysis of paired PBMC samples, we provide an in-depth characterization of the three main immunological compartments of grade 4 glioma.

Previous studies focused on the description of the TME of grade 4 glioma, which also considered regional differences, yet they focused primarily at neoplastic cells rather than the immune compartment (*Darmanis et al., 2017*). Others investigated the differences in the iTME composition between primary and metastatic brain tumors (*Klemm et al., 2020*; *Friebel et al., 2020*). Interestingly, the two latter ones reported differences in the iTME composition between $IDH1^{wt}$ and $IDH1^{mut}$ glioma. Of note, both authors included low-grade and even pre-treated recurrent glioma patients into the $IDH1^{mut}$ group, representing a quite heterogenous patient cohort. In this study, we aimed at providing a representative selection of primary, treatment-naïve grade 4 glioma patients including $IDH1^{wt}$ and $IDH1^{mut}$ to identify common transcriptional differences within the iTME between tumor center, periphery and PBMC of grade 4 glioma. How far these regional differences vary between $IDH1^{wt}$ and $IDH1^{mut}$ grade 4 glioma merits further investigation.

We identified a transcriptionally distinct MG subcluster, MG_1, which was enriched in the infiltrative tumor periphery and displayed an anti-inflammatory/non-reactive phenotype. A similar MG subpopulation expressing a comparable gene signature was recently described to be enriched in Alzheimer's disease patients (*Olah et al., 2020*).

Additionally, the peripheral cytotoxic cell compartment exhibited an impaired activation state, including a downregulated IFN response signature in CD8+ T cells. Induction of an IFN response state was described as a consequence of T cell receptor-mediated IFN-γ production, likely serving as an autocrine response and inducing the proliferative program (*Szabo et al., 2019*). Hence, the reduced autocrine IFN-responsive state in the tumor peripheral CD8+ T cells, together with downregulated proliferative and co-stimulatory genes emphasized their impaired activation in the peripheral infiltration zone. Recently, we showed that the response to immunotherapy in GBM is indeed region dependent. For this, we cultured GBM explants in perfusion bioreactors and treated with anti-CD47, anti-PD1, or their combination which induced an IFN-γ response only in the tumor center, but not periphery (*Shekarian et al., 2022*). Adding experimental support to the here described impaired activation signature in the tumor periphery.

By exploring the transcriptional trajectory of CD8+ T cells from the blood circulation into the immunosuppressive TME of the tumor periphery, we uncovered CX3CR1^high and CX3CR1^int effector and memory CD8+ T cells, respectively, to be highly enriched in the PBMC, but absent in the iTME. Recently, adoptive transfer studies of CX3CR1+ CD8+ T cells in a melanoma mouse model significantly suppressed tumor growth (*Yan et al., 2018*). Others identified increased frequencies of CX3CR1+ CD8+ T cells in non-small cell lung and melanoma patients who responded to anti-PD-1 therapy, where these cells exhibited migratory capabilities into the tumor site followed by potent tumor rejection (*Yamauchi et al., 2021*; *Yan et al., 2018*). Thus, the authors proposed T cell CX3CR1 expression as a predictor of response to ICI therapy. Even though CX3CR1+ CD8+ T cells might not be specific to GBM, as they are found in healthy donor and different inflammation related conditions (*Böttcher et al., 2015*; *Gerlach et al., 2016*; *Yamauchi et al., 2021*; *Yan et al., 2018*), the absence of these effector and potentially ICI therapy responsive CD8+ T cells in the glioma TME could serve as an additional explanation for the disappointing outcome of clinical trials using ICI in glioma patients.

The observed $T_{rm}$ exhaustion phenotype of the glioma residing CD8+ T cells was recently reported as well for tumor-infiltrating PD-1^high CD8+ T cells in hepatocellular carcinoma (*Ma et al., 2019*). Whether these glioma-associated CD8+ T cells really possess tumor-specificity requires further study. Particularly in the light of a recent study by Smolders and colleagues who reported a consistent brain-resident CD8+ T cell population in a miscellaneous autopsy cohort of patients with neurological disorders excluding brain malignancies (Alzheimer's disease, Parkinson's disease, dementia, depression, multiple sclerosis), as well as patients with no known brain disease. These brain-resident CD8+ T cells displayed a remarkably consistent $T_{rm}$ phenotype (*Smolders et al., 2018*). The authors further showed high expression of inhibitory receptors CTLA-4 and PD-1 on the brain-resident CD8+ $T_{rm}$ cells, which is in line with the core phenotypic signature of $T_{rm}$ cells from other tissues (*Kumar et al., 2017*; *Mackay et al., 2013*). Yet, the brain CD8+ $T_{rm}$ cells showed a preserved inflammatory potential with substantial production of IFN-γ and TNF-α upon ex vivo stimulation. They concluded that extensive immune activation with release of highly neurotoxic lytic enzymes, such as perforin and granzyme B, harmfully impacts the brain parenchyma and should be tightly controlled, whilst maintaining the capability to

elicit a fast inflammatory response when a neurotropic virus threatens the CNS (*Smolders et al., 2018*). Therefore, inhibitory receptors like PD-1 and CTLA-4 on brain CD8[+] T$_{rm}$ cells may support CNS homeostasis by preventing uncontrolled T cell reactivity, and the availability of the receptor ligands may determine their inhibitory effect. While this may represent a well-balanced equilibrium under healthy conditions, the tumor setting leads to its disruption with upregulation of inhibitory ligands like PD-L1 on glioma cells and CD86 on GAMs, leading to the dysfunctional state seen in the glioma-residing CD8[+] T$_{rm}$ cells.

Another study comprehensively showed, that peripheral infections generate antigen-specific CD8[+] T$_{rm}$ cells in the brain, mediating protection against CNS infections (*Urban et al., 2020*). These data could implicate that the glioma-associated CD8[+] T cells are devoid of tumor-specific reactivity, but rather represent a pre-existing T cell population generated after peripheral infections, which acquired a dysfunctional state upon glioma formation. To test this hypothesis, further characterization of these cells is required, including analysis of T cell receptor clonality and tumor-specificity by patient-matched T cell/glioma-sphere co-culture assays.

Lastly, our cell-cell interaction analysis revealed signaling pathways between peripheral MG and lymphocytes potentially inducing the observed impaired activation signature. The predicted interaction between MG SPP1 and NK cells integrin complex *ITGA4-ITGB1* (CD49d-CD29), might mediate NK cell adhesion and migration (*Gandoglia et al., 2017*). This may facilitate interaction of inhibitory NK receptors KLRB1 and CD94/NKG2A with MG C-type lectin-related ligands and HLA-E, respectively, which could explain the observed impaired activation state of peripheral NK cells. Moreover, SPP1/CD44 interaction in T lymphocytes was described to suppress cell activation and proliferation (*Klement et al., 2018*). In a comprehensive approach where transcriptional states of human MG were mapped during aging and disease, *SPP1* was found to be differentially expressed in aging-microglia. This was associated to a doubling of the abundance of SPP1[+] GAMs in GBM samples compared to MG from age-matched controls (*Sankowski et al., 2019*). These observations support a possible role of MG SPP1 in glioma progression. More recently, myeloid-derived osteopontin (encoded by *SPP1*) was shown to trigger a chronic activation of NFAT2 in tumor-reactive CD8[+] T cells, leading to T cell dysfunction and exhaustion in an experimental mouse model of GBM (*Kilian et al., 2023*). This adds an additional layer of mechanistic evidence for the *SPP1*/Osteopontin-mediated signaling axis leading to CD8[+] T cell dysfunction in GBM.

Limitations of our study include the limited patient number, thereby our study was neither designed nor powered to explore differences in neoplastic cells, given the high inter- and intra-patient variability in glioma cells (*Darmanis et al., 2017*). Importantly, our dataset establishes a starting point for further interrogation and provides a first analysis of the transcriptional landscape of the major immune populations in grade 4 glioma within three important regional compartments. Further, we confirmed the observed phenotype of CD8[+] T cells in the blood and tumor periphery by flow cytometry in a cohort of ten additional patients, addressing possible generalization concerns. Together, we provide a regionally resolved map of leukocyte activation in the TME and blood circulation from grade 4 glioma patients, helping the research community to uncover novel therapeutic strategies to combat this fatal disease.

# Methods

**Key resources table**

| Reagent type (species) or resource | Designation | Source or reference | Identifiers | Additional information |
| --- | --- | --- | --- | --- |
| Biological sample (Human adult GBM tissue samples) | Tumor Center; Tumor Periphery | Neurosurgical Clinic of the University Hospital of Basel, Switzerland | | |
| Biological sample (Human Peripheral Blood Buffy Coat) | PBMC | Neurosurgical Clinic of the University Hospital of Basel, Switzerland | | |
| Antibody | Rat monoclonal anti-human/mouse CD11b (clone M1/70), FITC | BioLegend | Cat# 101206 | FACS: 1:50 |

*Continued on next page*

*Continued*

| Reagent type (species) or resource | Designation | Source or reference | Identifiers | Additional information |
|---|---|---|---|---|
| Antibody | Mouse monoclonal anti-human CD45 (clone 2D1), FITC | BioLegend | Cat# 368508 | FACS: 1:50 |
| Antibody | Fc-Block anti-human CD16/CD32, TruStain FcX | BioLegend | Cat# 422302 | FACS: 1:50 |
| Antibody | Mouse monoclonal anti-human CD45RO (clone UCHL1), APC | BioLegend | Cat# 304210 | FACS: 1:25 |
| Antibody | Mouse monoclonal anti-human CD45RA (clone HI100), PE | BioLegend | Cat# 304108 | FACS: 1:25 |
| Antibody | Mouse monoclonal anti-human CD3e (clone UCHT1), BV650 | BioLegend | Cat# 300468 | FACS: 1:25 |
| Antibody | Mouse monoclonal anti-human CD8a (clone RPA-T8), BV421 | BioLegend | Cat# 301036 | FACS: 1:25 |
| Antibody | Mouse monoclonal anti-human CCR7 (clone G043H7), FITC | BioLegend | Cat# 353216 | FACS: 1:25 |
| Antibody | Mouse monoclonal anti-human CD62L (clone DREG-56), AF700 | BioLegend | Cat# 304820 | FACS: 1:50 |
| Antibody | Mouse monoclonal anti-human CD69 (clone FN50), APC/Cy7 | BioLegend | Cat# 310914 | FACS: 1:25 |
| Antibody | Mouse monoclonal anti-human CD103 (clone Ber-ACT8), PerCP/Cy5.5 | BioLegend | Cat# 350226 | FACS: 1:25 |
| Antibody | Rat monoclonal anti-human CX3CR1 (clone 2A9-1), PE-Cy7 | BioLegend | Cat# 341612 | FACS: 1:25 |
| Commercial assay or kit | Chromium Single Cell 3' Reagent Kits v3 | 10 x Genomics | Cat# CG000183 | |
| Commercial assay or kit | BioAnalyzer High Sensitivity DNA Analysis kit | Agilent | Cat# 5067–4626 | |
| Commercial assay or kit | Qubit dsDNA High Sensitiivity assay kit | ThermoFisher | Cat# Q33230 | |
| Chemical compound, drug | Collagenase-4 | Worthington Biochemical Cooperation | Cat# LS004188 | |
| Chemical compound, drug | DNAse1 | Roche | Cat# 10104159001 | |
| Chemical compound, drug | ACK lysis buffer | Gibco | Cat# A1049201 | |
| Chemical compound, drug | Bambanker | Nippon Genetics | Cat# BB01 | |
| Chemical compound, drug | Sucorse | Sigma Aldrich | Cat# 84100 | |
| Chemical compound, drug | Ficoll-Paque PLUS | Cytiva | Cat# 17144002 | |
| Chemical compound, drug | Live/Dead Fixable Near IR Dead Stain Kit, APC-Cy7 | ThermoFisher | Cat# L34976 | |
| Chemical compound, drug | Zombie Aqua Fixable Viability Kit | BioLegend | Cat# 423102 | |
| Software, algorithm | R environment, version 4.1 | R Core Team | https://www.r-project.org/ | |
| Software, algorithm | GraphPad Prism 9 | GraphPad Software Inc. | N/A | |
| Software, algorithm | Flow Jo, version 10.8.1 | Tree Star | N/A | |

## Glioma tissue dissociation

Resected glioma tissue samples were immediately placed on ice and transferred to the laboratory for single cell dissociation within 2–3 hr after resection. Human brain tissue was manually minced using razor blades and enzymatically dissociated at 37 °C for 30 min with 1 mg/ml collagenase-4

(#LS004188, Worthington Biochemical Corporation, USA) and 250 U/ml DNAse1 (#10104159001, Roche, Switzerland) in a buffer containing Hank's Balanced Salt Solution (HBSS) with $Ca^{2+}/Mg^{2+}$, 1% MEM non-essential amino acids (Gibco, USA), 1 mM sodium pyruvate (Gibco), 44 mM sodium bi-carbonate (Gibco), 25 mM HEPES (Gibco), 1% GlutaMAX (Gibco) and 1% antibiotic-antimycotic (Sigma-Aldrich, USA). Cells were filtered and separated from dead cells, debris and myelin by a 0.9 M sucrose (#84100, Sigma-Aldrich) density gradient centrifugation. Upon ACK-lysis for removal of erythrocytes (#A1049201, Gibco) the now generated single-cell suspension (SCS) was washed, counted and frozen in Bambanker (#BB01, Nippon Genetics, Germany) in liquid nitrogen until use.

## PBMCs (peripheral blood mononuclear cells) preparation

Patient blood samples were directly placed on ice and transferred to the laboratory for PBMC isolation. Blood samples were centrifuged to separate buffy coat from plasma and erythrocytes, followed by standard density gradient centrifugation protocol (#17144002, Ficoll-Paque PLUS, Cytiva, USA) to isolate PBMCs. PBMCs were frozen in Bambanker (#BB01, Nippon Genetics, Germany) in liquid nitrogen until use.

## FACS sorting for single-cell RNA sequencing (scRNA-seq)

Cryopreserved tumor digests from glioma samples (center and periphery), as well as autologous PBMCs were thawed and washed with excess ice-cold 1xPBS and spun down at 350 x $g$ for 5 min. Subsequently, the cells were stained with Live/Dead (APC-Cy7 (Near IR), # L34976, Thermo Fischer) and a cocktail of fluorescently conjugated antibodies CD11b (FITC, clone M1/70, #101206, BioLegend) and CD45 (FITC, clone 2D1, #368508, BioLegend), and large debris were removed with a 40 μm strainer. All samples were acquired on the BD FACS ARIA Fusion III (Becton Dickinson GmbH, Germany). For single-cell RNA-seq experiments, live and single gated cells were sorted into non-immune cell (CD45$^-$ CD11b$^-$) and immune cell (CD45$^+$CD11b$^+$) populations. Both populations were directly sorted into Eppendorf tubes with 1xPBS supplemented with 1% BSA for single cell RNA sequencing.

## Single-cell RNA sequencing (scRNA-seq) – Library preparation and sequencing

Single-cell RNA-seq was performed using Chromium Single Cell 3' GEM, Library & Gel Bead Kit v3 (#CG000183, 10 x Genomics, Pleasanton, CA, USA) following the manufacturer's protocol. Briefly, non-immune cells and immune cells were mixed at a defined ratio of 1:4. Roughly 8000–10,000 cells per sample, diluted at a density of 100–800 cells/μL in PBS plus 1% BSA determined by Cellometer Auto 2000 Cell Viability Counter (Nexelom Bioscience, Lawrence, MA), and were loaded onto the chip. The quality and concentration of both cDNA and libraries were assessed using an Agilent BioAnalyzer with High Sensitivity kit (#5067–4626, Agilent, Santa Clara, CA USA) and Qubit Fluorometer with dsDNA HS assay kit (#Q33230, Thermo Fischer Scientific, Waltham, MA) according to the manufacturer's recommendation. For sequencing, samples were mixed in equimolar fashion and sequenced on an Illumina HiSeq 4000 with a targeted read depth of 50,000 reads/cell and sequencing parameters were set for Read1 (28 cycles), Index1 (8 cycles), and Read2 (91 cycles).

## Single-cell RNA sequencing (scRNA-seq) - Computational analysis

The dataset was analyzed by the Bioinformatics Core Facility, Department of Biomedicine, University of Basel. Read quality was controlled with the FastQC tool (version 0.11.5). Sequencing files were processed using the Salmon Alevin tool (v 1.3.0) (*Srivastava et al., 2019*) to perform quality control, sample demultiplexing, cell barcode processing, pseudo-alignment of cDNA reads to the human Gencode v35 reference and counting of UMIs. Parameters *--keepCBFraction 1* and *--maxNumBarcodes 100000* were used.

Processing of the UMI counts matrix was performed using the Bioconductor packages DropletUtils (version 1.8.0) (*Griffiths et al., 2018*; *Lun et al., 2019*), scran (version 1.16.0) (*Vallejos et al., 2017*; *Lun et al., 2016*) and scater (version 1.16.2) (*McCarthy et al., 2017*), following mostly the steps illustrated in the OSCA book (http://bioconductor.org/books/release/OSCA/) (*Lun et al., 2016*; *Amezquita et al., 2020*). Filtering for high-quality cells was done based on library size (at least 2000 UMI counts per cell), the number of detected genes (at least 700 genes detected) and the percentage of reads mapping to mitochondrial genes (larger than 0% and lower than 15%), based on the distribution

observed across cells. Low-abundance genes with average counts per cell lower than 0.006 were filtered out. The presence of doublet cells was investigated with the scDblFinder package (version 1.2.0), and suspicious cells were filtered out (score >0.6). After quality filtering, the resulting dataset consisted of UMI counts for 15,523 genes and 45,466 cells, ranging from 803 to 9,121 per sample.

UMI counts were normalized with size factors estimated from pools of cells created with the scran package *quickCluster*() function (*Vallejos et al., 2017*; *Lun et al., 2016*). To distinguish between genuine biological variability and technical noise we modeled the variance of the log-expression across genes using a Poisson-based mean-variance trend. The scran package *denoisePCA*() function was used to denoise log-expression data by removing principal components corresponding to technical noise. Consistent to the *findElbowPoint*() function from the PCAtools Bioconductor package, this led us to retain the top 5 principal components (PCs), explaining 48.3% of the total variance in the dataset, for the clustering and dimensionality reduction steps.

Since the excluded deeper PCs (PC8 and 10) were associated to patient-specific effects (which in our experimental setup are also confounded with batch effects), this choice dispensed us from performing an additional correction for patient-specific effects. A quantitative test of overlap of cells across patients was made using the CellMixS package (*Lütge et al., 2021*) (version 1.12.0) developed to quantify the effectiveness of batch correction methods. We used the cell-specific mixing (CMS) score, which highlighted a very good overlap across cells from different patients in the lymphoid compartment and for the monocytes. The myeloid compartment displayed a slightly elevated patient-specific structure, while it was most pronounced for the CD45 negative subset.

A t-stochastic neighbor embedding (t-SNE) was built with a perplexity of 50 using the top most variable genes (141 genes with estimated biological variance >0.3, excluding genes with highest proportion of reads in the ambient RNA pool estimated from empty droplets), and the denoised principal components as input. Clustering of cells was performed with hierarchical clustering on the Euclidean distances between cells (with Ward's criterion to minimize the total variance within each cluster *Murtagh and Legendre, 2014*; package cluster version 2.1.0). The number of clusters used for following analyses was identified by applying a dynamic tree cut (package dynamicTreeCut, version 1.63–1) (*Langfelder et al., 2008*), resulting in 10 with argument *deepSplit* set to 1, or 22 clusters with argument *deepSplit* set to 2. This clustering was validated with an alternative clustering approach using a graph-based approach and Louvain algorithm for community detection (using the *FindNeighbors*() and *FindClusters*() functions from the *Seurat* package (*Hao et al., 2021*) (version 4.3.0), with a resolution of 0.7) which showed a good correspondence to our hierarchical clustering.

The *findMarkers* function of the scran package was used to find the best markers across annotated cell types (parameters *direction="up"* and *pval.type="any"*). The top 10 markers for each cell type were extracted and pooled to from a list of 68 markers.

The Bioconductor package *SingleR* (version 1.2.4) was used for cell-type annotation of the cells (*Aran et al., 2019*) using as references (i) a public bulk RNA-seq dataset of sorted immune cell types from human PBMC samples (*Monaco et al., 2019*), available through the *celldex* Bioconductor package; (ii) a bulk RNA-seq dataset of sorted immune cell types from the tumor microenvironment of human gliomas (*Klemm et al., 2020*) (UMI count matrix and annotation downloaded from https://joycelab.shinyapps.io/braintime/); (iii) a 10 X genomics scRNA-seq dataset of TAMs from the tumor micro-environment of glioblastoma tumors from seven newly diagnosed human patients *Pombo Antunes et al., 2021*; (iv) a microglia and a macrophage signature scores were defined by averaging the center and scaled expression levels of gene lists obtained in *Müller et al., 2017* and (v) a Smartseq2 scRNA-seq dataset of IDH-wild-type glioblastoma tumors (*Neftel et al., 2019*) (downloaded from GEO accession GSE131928). An endothelial score was defined by averaging the center and scaled expression levels of the genes *CDH5, VWF, CD34,* and *PECAM1*.

The SingleR high-quality assignments (pruned scores) from the comparisons to the multiple references (some being better at enlightening some immune subsets than others) were used to manually derive a consensus cell type annotation for each cluster. This manual annotation was done jointly with the input from the signature scores and relative expression patterns of known marker genes and of cluster-specific genes.

When we judged that the resolution was not sufficient to discriminate well the subtypes of cells within a subset (in particular for the CD45neg and myeloid cells subsets), we isolated these cells

and re-performed selection of hyper-variable genes, dimensionality reduction and clustering on the isolated subset.

Differential abundance analysis of the identified cell types between tumor sites was performed using *limma-voom* (*Law et al., 2014*). This method, implemented in the *diffcyt* package, was shown to be able to handle well the overdispersion in the cell type proportions estimates typically observed with single cell technologies (*Weber et al., 2019*). Differential abundance of cell types was considered to be significant at a false discovery rate (FDR) lower than 5%. To provide increased resolution into local trends within the tumor innate immune subset we used the *miloR* bioconductor package (*Dann et al., 2022*; version 1.4.0) to test for differential abundance on neighborhoods of a *k*-nearest neighbor graph. Testing was done with *limma-voom* similarly to above. Cell type labels were assigned to each neighborhood by finding the most abundant cell type across cells and requiring that at least 70% of the cells come from one cell type.

Differential expression between tumor sites, or between PBMC cells and tumor periphery cells, stratified by annotated cell type, was performed using a pseudo-bulk approach, which was shown to be the best-performing approach for differential expression testing in recent benchmarks (*Junttila et al., 2022*). UMI counts were summed across cells from each cell type in each sample when at least 20 cells could be aggregated. The aggregated samples were then treated as bulk RNA-seq samples (*Lun and Marioni, 2017*) and for each pairwise comparison genes were filtered to keep genes detected in at least 5% of the cells aggregated. The package edgeR (version 3.30.3; *Robinson et al., 2010*) was used to perform TMM normalization (*Robinson and Oshlack, 2010*) and to test for differential expression with the Generalized Linear Model (GLM) framework, using a model accounting for patient-specific effects. Genes with a FDR lower than 5% were considered differentially expressed. To validate our differential expression analysis, we rerun a pseudo-bulk expression analysis using DESeq2 (version 1.38.3; *Love et al., 2014*) for the main immune populations in the periphery versus center and periphery versus PBMC comparison. Although the results were not identical, they overall agreed with edgeR (*Supplementary file 7*). Gene set enrichment analysis was performed with the function camera (*Wu and Smyth, 2012*) on gene sets from the Molecular Signature Database (MSigDB, version 7.4; *Liberzon et al., 2015*; *Subramanian et al., 2005*). We retained only sets containing more than five genes, and gene sets with a FDR lower than 5% were considered as significant.

## Cell chat analysis

The R package CellChat (1.1.3) (*Jin et al., 2021*) was used to analyze cell-cell interactions in our dataset (with previously annotated nine cell types). We followed the recommended workflow to infer the cell state-specific communications (using *identifyOverExpressedGenes*, *identifyOverExpressedInteractions* and *projectData* with the default parameters). We performed three separate analyses, on the center and the periphery subsets and a comparison analysis as described in the official workflow. We visualized the significant interactions for the microglia cluster using *netVisual_chord_gene* and used *plotGeneExpression* to display of the expression of all genes involved SPP1 signaling pathway in the cell populations. Finally, *netAnalysis_signalingRole_scatter* was used to calculate and visualize incoming and outgoing signaling strength.

## Flow cytometry analysis of paired PBMC and periphery samples

Cryopreserved samples were thawed and washed with excess ice-cold 1xPBS and spun down at 350 x *g* for 5 min. Cells were resuspended in FACS buffer (PBS plus 2% FBS) and blocked with monoclonal antibody to CD16/32 (Human TruStain FcX, #422302, Biolegend) for 10 min at 4 °C before staining with surface antibodies: CD45RA (PE, clone HI100, #304108), CD45RO (APC, clone UCHL1, #304210), CD3e (BV650, clone UCHT1, #300468), CD8a (BV421, clone RPA-T8, #301036), CCR7 (FITC, clone G043H7, #353216), CD62L (AF700, clone DREG-56, #304820), CD69 (APC-Cy7, clone FN50, #310914), CD103 (PerCP/Cy5.5, clone Ber-ACT8, #350226) and CX3CR1 (PE/Cy7, clone 2A9-1, #341612). All antibodies were purchased from BioLegend, USA. Cells were stained for 30 min at 4 °C, and subsequently washed with FACS buffer. To exclude dead cells Zombie Aqua Fixable Viability Kit (#423102, 1:100, BioLegend) was added. Acquisition was performed on a CytoFLEX (Beckman). Data was analyzed using FlowJo software, version 10.8.1 (TreeStar). Gates were drawn by using Fluorescent Minus One (FMO) controls.

## Statistical analysis of flow cytometry data

Data analysis and graph generation was performed using GraphPad Prism 9 (GraphPad Prism Software Inc). Paired comparisons between two groups were performed using Wilcoxon matched-pairs signed rank test. Differences of more than two paired groups were assessed using repeated measures one-way ANOVA test, followed by post-hoc Šidák's multiple comparisons correction. A p-value <0.05 was considered statistically significant. *p≤0.05, **p≤0.01, ***p≤0.001, ****p≤0.0001.

## Graphical illustrations

All graphical illustrations were created with BioRender.com.

## Acknowledgements

We are grateful to the patients and their families for their consent to donate tissue to our brain tumor biobank. We thank Tamara Hüssen, Alison Riberio and Florian Limani for experimental support and Cecile Buenter for critical revision of the manuscript. Calculations were performed at sciCORE (http://scicore.unibas.ch/) scientific computing center at the University of Basel.

## Additional information

### Competing interests

Steffen Dettling: S.D. is affiliated with Roche Pharmaceutical Research and Early Development. The author has no financial interests to declare. Sylvia Herter: S.H. is affiliated with Roche Pharmaceutical Research and Early Development. The author has no financial interests to declare. Marina Bacac: M.B. is affiliated with Roche Pharmaceutical Research and Early Development. The author has no financial interests to declare. Gregor Hutter: G.H. has equity in, and is a cofounder of Incephalo Inc. The other authors declare that no competing interests exist.

### Funding

| Funder | Grant reference number | Author |
|---|---|---|
| Swiss Cancer Research Foundation | MD-PhD-4818-06-2019 | Philip Schmassmann |
| Schweizerischer Nationalfonds zur Förderung der Wissenschaftlichen Forschung | PP00P3_176974 | Gregor Hutter |
| Swiss Cancer Research Foundation | KFS- 4382-02-2018 | Gregor Hutter |
| Brain Tumour Charity | GN- 000562 | Gregor Hutter |
| Department of Surgery, University Hospital Basel | | Philip Schmassmann |

The funders had no role in study design, data collection and interpretation, or the decision to submit the work for publication.

### Author contributions

Philip Schmassmann, Conceptualization, Data curation, Validation, Visualization, Methodology, Writing – original draft, Writing – review and editing; Julien Roux, Data curation, Formal analysis; Steffen Dettling, Data curation, Methodology; Sabrina Hogan, Data curation, Formal analysis, Methodology; Tala Shekarian, Tomás A Martins, Marie-Françoise Ritz, Writing – review and editing; Sylvia Herter, Conceptualization, Funding acquisition, Project administration; Marina Bacac, Funding acquisition, Project administration; Gregor Hutter, Conceptualization, Resources, Supervision, Funding acquisition, Project administration, Writing – review and editing

## Author ORCIDs

Philip Schmassmann ![ORCID] https://orcid.org/0000-0002-9411-3852
Julien Roux ![ORCID] https://orcid.org/0000-0002-4192-5099
Sabrina Hogan ![ORCID] http://orcid.org/0000-0003-3994-8109
Tomás A Martins ![ORCID] https://orcid.org/0000-0001-6016-2288
Marie-Françoise Ritz ![ORCID] http://orcid.org/0000-0002-5650-9987

## Ethics

Human adult GBM tissue samples were obtained at the Neurosurgical Clinic of the University Hospital of Basel, Switzerland in accordance with the Swiss Human Research Act and institutional ethics commission (EKNZ 02019-02358). All patients gave written informed consent for tumor biopsy collection and signed a declaration permitting the use of their biopsy specimens in scientific research, including storage in our brain tumor biobank (Req-2019-00553). All patient identifying information was removed and tissue was coded for identification.

Reviewer #1 (Public Review): https://doi.org/10.7554/eLife.92678.2.sa1
Reviewer #2 (Public Review): https://doi.org/10.7554/eLife.92678.2.sa2

---

# Additional files

## Supplementary files

- Supplementary file 1. Patient characteristics.
- Supplementary file 2. QC of scRNA-seq data.
- Supplementary file 3. Cell count per cell cluster, site and patient.
- Supplementary file 4. Differentially expressed genes per cell cluster in Periphery versus Center.
- Supplementary file 5. Differentially expressed genes and differential abundance testing of MG subclusters.
- Supplementary file 6. Differentially expressed genes per cell cluster in Periphery versus PBMC.
- Supplementary file 7. Pseudo-bulk expression analysis using DESeq2.
- MDAR checklist

## Data availability

The UMI count matrix and cell metadata from the scRNA-seq dataset are available on GEO under accession number GSE197543.

The following dataset was generated:

| Author(s) | Year | Dataset title | Dataset URL | Database and Identifier |
|---|---|---|---|---|
| Schmassmann P, Hutter G, Roux J | 2022 | Single-cell characterization of human glioblastoma reveals regional differences in tumor-infiltrating leukocyte activation | https://www.ncbi.nlm.nih.gov/geo/query/acc.cgi?acc=GSE197543 | NCBI Gene Expression Omnibus, GSE197543 |

The following previously published datasets were used:

| Author(s) | Year | Dataset title | Dataset URL | Database and Identifier |
|---|---|---|---|---|
| Monaco G, Lee B, Xu W, Hwang Y, Poidinger M, Poidinger M, de Magalhães J, Larbi A | 2019 | RNA-Seq profiling of 29 immune cell types and peripheral blood mononuclear cells | https://www.ncbi.nlm.nih.gov/geo/query/acc.cgi?acc=GSE107011 | NCBI Gene Expression Omnibus, GSE107011 |

*Continued on next page*

*Continued*

| Author(s) | Year | Dataset title | Dataset URL | Database and Identifier |
|---|---|---|---|---|
| Klemm F, Maas RR, Bowman RL, Kornete M, Soukup K, Nassiri S, Brouland J-P, Iacobuzio-Donahu CA, Brennan C, Tabar V, Gutin PH, Daniel RT, Hegi ME, Joyce JA | 2020 | Interrogation of the Microenvironmental Landscape in Brain Tumors Reveals Disease-Specific Alterations of Immune Cells | https://joycelab.shinyapps.io/braintime/ | Joyce Lab Brain Tumor Immune Micro Environment, Klemm et al., Cell2020 |
| Antunes AR | 2020 | Single-cell profiling of myeloid cells in glioblastoma across species and disease stage reveals macrophage competition and specialization | https://www.ncbi.nlm.nih.gov/geo/query/acc.cgi?acc=GSE163120 | NCBI Gene Expression Omnibus, GSE163120 |
| Müller et al. | 2017 | Single-cell profiling maps the spectrum of crosstalk between glioma cells and tumour associated macrophages | https://www.omicsdi.org/dataset/ega/EGAS00001002185 | OmicsDI, EGAS00001002185 |
| Neftel et al. | 2019 | Single cell RNA-seq analysis of adult and paediatric IDH-wildtype Glioblastomas | https://www.ncbi.nlm.nih.gov/geo/query/acc.cgi=GSE131928 | NCBI Gene Expression Omnibus, GSE131928 |

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
