## [Editor Report · eLife assessment]

This study is **valuable** and contains results that are supported by **convincing** evidence. In the future, the observations could be further strengthened by independent validation, and by looking at larger numbers of patients, as well as by determining whether patient heterogeneity is either contributing to or obscuring certain patterns. The work will be of interest to a broad audience in the oncology and immunology fields as it is on a cancer type that does not respond well to immune checkpoint therapeutics.

---

## [Referee Report · Reviewer #1 (Public Review)]

Summary:

In this manuscript, Schmassmann et al. present a study on the immune microenvironment of grade 4 gliomas using single-cell RNA-seq data from the tumor center, periphery, and peripheral blood of patients. This manuscript is overall well written and reads easily. The approach to studying the TME at various spatial locations is innovative and interesting, and the dataset presented has the potential to become a useful resource for the community. However, the size of the dataset, notably in the context of the important inter-patient variability on key clinical information, hinders the generalizability of the results. The analysis presented by the authors seems at times somewhat shallow as compared to other studies in the literature, being almost solely based on the analysis of a single dataset with extremely limited biological validation of the observations, and some claims made by the authors do not seem appropriately backed by the data they present. While I appreciate the vast analysis effort undertaken by the authors, it seems more work is required to make the most of this interesting dataset and substantiate the conclusions.

Strengths:

The authors have provided useful insights into diverse GBMs (IDH mutant and IDH wild-type) that provide a deep assessment of individual tumors with spatial information.

Weaknesses:

A larger set of tumors will need to be explored before general principles of immune biology and GBM immune evasion can be uncovered. This is a descriptive study that provides some interesting new hypotheses - but these will need deeper functional exploration.

---

## [Referee Report · Reviewer #2 (Public Review)]

Summary:

Most of this paper concerns scRNA-seq data generated from glioblastoma patients, from three regions: tumor center, tumor periphery, and peripheral blood mononuclear cells. They focus on immune cells, and especially microglia and T-cells, where they look at the presence/absence/changes in different types of immune signatures. The data and analysis are sound and supportive of the conclusions they draw, though future studies with more patients and/or low-throughput validation would strengthen their evidence. This study adds to our knowledge of the immune cell environment in glioblastoma patients and its regional variation.

Strengths:

A key strength of the paper lies in the novelty of the data, which simultaneously examines, at single-cell resolution, gene expression in two different tumor regions (center and periphery) along with peripheral blood. The authors provide numerous detailed and state-of-the-art analyses of this data, including gene differential expression, differential abundance of cell types, gene ontology analyses, tSNE visualizations, etc.

They focus in particular on differences in immune cell types. There are some suggestive differences in immune cell composition of center versus periphery, although the number of patients (5, one of whom is missing center data) does not allow one to draw a definitive conclusion.

They identified more definitive gene expression differences in center versus peripheral microglia -- differences that were not reflected in other cell types, and which included downregulation of a number of immune response functions. They also identified gene expression differences between two subsets of microglia, although those may partly reflect regional differences (the subsets are differentially enriched in the center versus periphery) or differential representation of different patients.

Finally, they identify differences in CD8+ T cells and NK cells in the center versus the periphery, where the latter were less activated/proliferative/cytotoxic.

Data analysis is performed to a high standard, using best-available methods and in some cases backed up with alternative approaches showing similar results.

Weaknesses:

While the nature of the dataset is novel, the relatively low patient numbers (five) and patient diversity (e.g. with regard to IHD1 status) may be obscuring differences in cell type abundances or cell state between regions.

Most discoveries based on the scRNA-seq discussed in the paper remain to be validated by low-throughput methods in either the same patient samples, if material remains, or in other patients.